# Fecal DNA metabarcoding helps characterize the Canada jay's diet and confirms its reliance on stored food for winter survival and breeding

**Alex O. Sutton**[1,2]*, **Dan Strickland**[3], **Jacob Lachapelle**[1], **Robert G. Young**[1], **Robert Hanner**[1], **Daniel F. Brunton**[4], **Jeffrey H. Skevington**[5], **Nikole E. Freeman**[1,2,6], **D. Ryan Norris**[1]

1 Department of Integrative Biology, University of Guelph, Guelph, Ontario, Canada, 2 School of Natural Sciences, Bangor University, Bangor, Wales, United Kingdom, 3 110, 205 1st Street, Courtenay, British Columbia, Canada, 4 Beaty Centre for Species Discovery and Botany Section, Canadian Museum of Nature, Ottawa, Ontario, Canada, 5 Carleton University, Ottawa, Ontario, Canada, 6 Division of Biology, Kansas State University, Manhattan, Kansas, United States of America

* a.sutton@bangor.ac.uk, alexosutto@gmail.com

**Data Availability Statement:** All relevant data are within the paper and its Supporting information files.

## Abstract

Accurately determining the diet of wild animals can be challenging if food items are small, visible only briefly, or rendered visually unidentifiable in the digestive system. In some food caching species, an additional challenge is determining whether consumed diet items have been previously stored or are fresh. The Canada jay (*Perisoreus canadensis*) is a generalist resident of North American boreal and subalpine forests with anatomical and behavioural adaptations allowing it to make thousands of arboreal food caches in summer and fall that are presumably responsible for its high winter survival and late winter/early spring breeding. We used DNA fecal metabarcoding to obtain novel information on nestling diets and compiled a dataset of 662 published and unpublished direct observations or stomach contents identifications of natural foods consumed by Canada jays throughout the year. We then used detailed natural history information to make informed decisions on whether each item identified to species in the diets of winter adults and nestlings was best characterized as 'likely cached', 'likely fresh' (i.e., was available as a non-cached item when it appeared in a jay's feces or stomach), or 'either possible'. Of the 87 food items consumed by adults in the winter, 39% were classified as 'likely cached' and 6% were deemed to be 'likely fresh'. For nestlings, 29% of 125 food items identified to species were 'likely cached' and 38% were 'likely fresh'. Our results support both the indispensability of cached food for Canada jay winter survival and previous suggestions that cached food is important for late winter/early spring breeding. Our work highlights the value of combining metabarcoding, stomach contents analysis, and direct observations to determine the cached vs. non-cached origins of consumed food items and the identity of food caches, some of which could be especially vulnerable to degradation through climate change.

**Funding:** This study was financially supported by the Natural Sciences and Engineering Research Council of Canada (https://www.nserc-crsng.gc.ca) in the form of a grant received by DRN and NEF. This study was also financially supported by Ontario Parks (https://www.ontarioparks.ca) in the form of a grant received by AOS, DRN, and NEF. This study was also financially supported by Friends of Algonquin Park (https://www.algonquinpark.on.ca) in the form of a grant. This study was also financially supported by the Wildlife Conservation Society Canada (https://www.wcscanada.org) in the form of a grant received by AOS and NEF. This study was also financially supported by the Ministry of Natural Resources and Forestry (https://www.ontario.ca/page/ministry-natural-resources-and-forestry) in the form of a grant received by AOS, DRN and NEF, in addition to providing logistical support for this project. The funders had no additional role in study design, data collection and analysis, decision to publish, or preparation of the manuscript.

**Competing interests:** The authors have declared that no competing interests exist.

## Introduction

While diet is closely linked to growth and individual performance [1–3], several factors frequently limit our ability to identify what free-living animals are consuming [4]. It is often difficult to observe what is eaten or fed to young either because individual food items are too small to identify, have been manipulated in ways that limit the ability of an observer to identify a food item (e.g., multiple food items combined into a single food bolus), or are hidden by the cryptic behaviour of adults (e.g., nesting high in a tree, building a well-concealed nest). Some of these difficulties can be overcome through the identification and quantification of stomach contents but this approach suffers from limitations of its own, including the need to collect samples before they are digested [5–7], the requirement that animals must either be found dead or sacrificed to obtain samples, and the difficulty of identifying or quantifying soft-tissue foods such as fungi, vertebrate flesh, or even some plant material [8].

Technological advances have overcome at least a few problems associated with estimating diet. In some circumstances, high-speed photography and/or the use of drones have enhanced our ability to identify food items being consumed by animals [9] and stable isotope analysis can be used to quantify diet composition during the period of tissue growth [6, 10]. Significant limitations nevertheless persist. Videography, for example, will likely never provide the ability to identify arthropods being delivered in a compact bolus directly into the mouth of a nestling passerine and, unless the diet of an animal is simple (few potential sources), stable isotope analysis is typically restricted to identifying only major food groups (e.g., invertebrates vs. vertebrates vs. plants). The problems associated with food item identification can be further complicated in species that cache food [11]. In these cases, it may not always be clear whether individuals are consuming food that is currently available (i.e., fresh) or food items that were stored weeks or months earlier, raising the possibility that a completely different suite of food resources had been available compared to when the food was consumed.

All these challenges have a bearing on attempts to elucidate the diet of the Canada jay (*Perisoreus canadensis*), an iconic, sedentary, food-caching species of North American boreal and subalpine forests. Canada jays regularly store food in trees, well above eventual snow levels, using sticky saliva from unusually large salivary glands [12, 13] to fasten individual food items under bark scales or tufts of lichens. Food storage is conspicuous late-summer and fall behaviour in Canada jays and, from the successful retrieval of stored food, presumably contributes to high (> 90%) adult winter survival, at least at the southern edge of their range [14, 15]. However, despite the likelihood that the recovery of cached food explains high winter survival and territorial fidelity of Canada jays, we have very little data on what sustains them throughout the winter and what proportion of this food originates from cached stores. A further reason why such information is of particular interest is that the main foods Canada jays are known to consume (arthropods; berries; fleshy fungi; and vertebrate flesh; [22]) are all highly perishable. It seems questionable that such foods could reliably retain sufficient nutritional value from summer storage to the onset of sub-freezing (degradation-arresting) temperatures in the fall to account for the high winter survival of Canada jays [14]. Strickland et al. [16] provided some experimental evidence that volatile anti-microbial resins of conifers, particularly spruce (*Picea* spp.), may account for reduced degradation of perishable Canada jay caches but, ideally, one would like some assurance that high winter jay survival is not attributable instead to some unsuspected non-perishable food, as in nutcrackers (*Nucifraga* spp.; [17]).

Unequivocally characterizing Canada jay diet and determining the importance of stored food is even more difficult in the breeding season. With no obvious sources of fresh food, Canada jay females form and lay clutches as early as mid-February in Algonquin Park, Ontario [18] and raise offspring to fledging by late April-early May when, at least historically, the

ground may still be snow covered, lakes are frozen, green-up has barely begun, and fewer than 10% of migratory passerines have returned let alone begun nesting themselves [19]. Experimental evidence supports the inference that stored food facilitates early breeding in Canada jays [20], but concrete evidence in the form of positively identified natural food items that could only have been stored at the time they were fed to nestlings has been lacking.

We had three primary aims in this study. First, we used dietary DNA (dDNA) metabarcoding [21] to expand our knowledge of the diet of nestling Canada jays. Second, using all published and non-published sources, we characterized, as completely as possible, nestling and adult diets of the Canada jay, identifying any differences between them and, for adults, any seasonal differences (i.e., winter versus "non-winter"). Third, we assessed the extent to which items identified in diets of nestlings and winter adults had likely been cached before being recovered by the adults and then consumed or fed to nestlings. In doing so, we also discuss the practicality and limitations of using fecal dDNA metabarcoding for determining previously difficult-to-obtain information on animal diets.

## Methods

### Study species and field site

Canada jays are year-round residents of North American boreal and subalpine forests [22], ranging from the tree line in the north to as far as Arizona in the south. We collected diet data as part of an ongoing long-term study begun in 1964 of Canada jay demography and behavior in Algonquin Provincial Park (APP) near the southern edge of its range in Ontario, Canada in the transition zone between the Great Lakes-St. Lawrence deciduous hardwood forest and the boreal forest [16]. Nest building begins in mid- to late-February with clutches initiated as early as February 22. Nestlings typically hatch in mid- to late-March under still wintry conditions and remain in the nest for approximately 23 days before fledging [22].

### Collection of nestling fecal samples for metabarcoding

Since the only information on the Canada Jay nestling diet was from the contents of ten nestling stomachs or castings summarized by Strickland and Ouellet [22], our first aim was to improve our understanding of the diet through fecal dDNA metabarcoding. From April and May 2015–2017, we collected 20 fecal sacs from Canada jay nestlings in APP. Fourteen fecal sacs were collected opportunistically while nestlings were removed from the nest to be banded when they were approximately 11 days old [1, 15]. Each fecal sac was collected from a unique individual, although six fecal sacs came from the same nest on the day that young fledged. In the latter case, because the individual that each fecal sac came from could not be determined with certainty, at least one nestling, likely more, was represented more than once. Cumulatively, fecal sacs came from 8 separate nests and at least 11 nestlings. Upon collection, fecal sacs were immediately stored in a 50mL falcon tube filled with 100% ethanol and stored in a -20˚C freezer within 12 hrs of being collected and until processing.

### DNA extraction, amplification, and sequencing

Three gene regions were targeted in the PCR amplifications. These regions included a 157 bp region of the animal barcode, a fragment of the Cytochrome *c* Oxidase I five prime region (COI-5P; primer set ZBJ-ArtF1c-ion and ZBJ-Art2-ion; [23]), a 350 bp region of the Internal Transcribed Spacer (ITS2) region for fungi (primer set ITS3-M13ion and ITS4-M13ion; [24]), and a 163 bp region of the Ribulose 1,5-Biphosphate Carboxylase (rbcLa) gene for plants (primer set rbcLaF-M13ion and MrbcL 163-R1-M13ion; [24, 25]).

DNA was extracted from fecal sacs for amplification following methods outlined by Prosser and Hebert [24]. This extraction included 400 $\mu$L lysis buffer 700 mM guanidine thiocyanate (Sigma), 30 mM EDTA pH 8.0 (Fisher Scientific), 30 mM Tris-HCl pH 8.0 (Sigma), 0.5% Triton X-100 (Sigma), 5% Tween-20 (Fluka Analytical) mixed with 2 mg/mL of Proteinase K (Promega). This lysis buffer was added to the fecal samples and incubated overnight at 56˚C with gentle shaking.

Purification of DNA followed [26], whereby lysate was mixed with two volumes of binding mix 3 M guanidine thiocyanate, 10 mM EDTA pH 8.0, 5 mM Tris-HCl pH 6.4, 2% Triton X-100, 50% ethanol and applied to a silica membrane spin column (Epoch Biolabs), 700 $\mu$L at a time, centrifuged at 6000g for 2 min, and repeated until no solution remained. After this step, the column was washed with 750 $\mu$L wash buffer (1.56 M guanidine thiocyanate, 5.2 mM EDTA pH 8.0, 2.6 mM Tris-HCl pH 6.4, 1.04% Triton X-100), 70% ethanol and centrifuged at 6000g for 2 min. The column was washed a second time with 750 $\mu$L of wash buffer (50 mM NaCl (Fisher Scientific), 0.5 mM EDTA pH 8.0, 10 mM Tris-HCl pH 7.4), 60% ethanol and centrifuged at 6000g for 4 min. The flow-through was discarded and the column was centrifuged at 10,000g for 4 min. The silica membrane was transferred to a clean 1.5 mL microfuge tube and dried at 56˚C for 30 min. To release DNA from the silica membrane, 50 $\mu$L elution buffer (10 mM Tris-HCl, pH 8.0, prewarmed to 56˚C) was added to the membrane and was left to incubate at room temperature for 1 min. The column was centrifuged at 10,000g for 5 min. to elute the DNA. The DNA was quantified using a Qubit 2.0 fluorometer (Life Technologies) and adjusted to approximately 0.5 ng/lL with elution buffer.

Next, a two-step polymerase chain reaction (PCR) amplification was conducted. The initial PCR amplified the DNA without the presence of Ion Torrent PGM (Life Technologies) multiplex identifier (MID) tags which provide identifiers for sequence reads to specific samples. This reaction had a total volume of 12.5 $\mu$L and was made up of 6.25 $\mu$L of 10% D-(+)-trehalose dihydrate (Fluka Analytical), 2.0 $\mu$L of Hyclone ultra-pure water (Thermo Scientific), 1.25 $\mu$L of 10X PlatinumTaq buffer (Invitrogen), 0.625 $\mu$L of 50 mM MgCl2 (Invitrogen), 0.125 $\mu$L of each 10 $\mu$M primer, 0.0625 $\mu$L of 10 mM dNTP (KAPA Biosystems), 0.060 $\mu$L of 5U/$\mu$L PlatinumTaq DNA Polymerase (Invitrogen), and 2 $\mu$L of template DNA. PCR thermocycler conditions for this reaction consisted of 94˚C for 5 min., 40–60 cycles (40 for ITS2, 60 for rbcLA and COI-5P) of 94˚C for 30 s, 48–53˚C for 30 s (53˚C for ITS2, 55˚C for rbcLa, 48˚C for COI-5P), 72˚C for 30–45 s (45 s for ITS2, 30 s for rbcLa and COI-5P), and a final extension of 72˚C for 10 min. The second PCR used the products from the first PCR and the same chemistry but included primers fusion primers (see Prosser and Hebert; [24]). The thermocycling conditions for this reaction were 94˚C for 4 min., 20–25 cycles (20 cycles for ITS2, 25 cycles for rbcLa and COI-5P) of 94˚C for 40 sec., 51–56˚C for 40 sec. (56˚C for ITS2, 55˚C for rbcLa, 51˚C for COI-5P), 72˚C for 30–45 sec. (45 sec. for ITS2, 30 sec. for rbcLa and COI-5P), with a final extension of 72˚C for 5 min. After amplification the products were cleaned using the magnetic bead protocol described in Prosser and Hebert [24] and then quantified using a Qubit 2.0 fluorometer and adjusted to 1 ng/$\mu$L. Cleaned and normalized products used for library construction following Prosser and Hebert [24] and were then sequenced unidirectionally on an Ion Torrent PGM using a 318 v.2 chip at the University of Guelph Centre for Biodiversity Genomics, following the manufacturer's instructions.

The choices of molecular markers for DNA metabarcoding followed Prosser and Hebert [24]. Three gene regions were chosen to provide information on Canada jay fungal (ITS2), botanical (rbcLa), and animal (COI-5P) diet components. The rbcLa region was selected as it is readily amplifiable across a large diversity of plant life and for its utility in placing specimens to family and/or genus taxonomic levels [27]. The ITS2 region was selected as it has been shown to provide specimen identifications to species for a diverse number of fungi [28]. The

COI-5P region has been selected as a "DNA barcode" standard region by the Consortium for the Barcode of Life and is well established as an effective barcode across all animal taxa of interest in this study [29, 30]. Selection of these three gene regions is further supported through the large number of publicly available sequence records against which we can compare our results (as opposed to other marker options).

## Bioinformatics

Sequencing data was demultiplexed using the gene region and MID tags. Each resulting data set (three gene regions for each sample) were analysed informatically by first removing primer and adapter sequences [31]; see Prosser and Hebert [24] for sequences used), removing reads shorter than 100 bp, removing reads with low quality scores (QV < 20; github.com/ucdavis-bioinformatics/sickle), and then by de-replicated reads with 100% identity (http://hannonlab.cshl.edu/fastx_toolkit/index.html). After these steps, each read was taxonomically assigned using BLAST and three databases: all flowering plant rbcLa sequences from BOLD, all insect COI-5P sequences from GenBank, and all fungi ITS2 sequences from BOLD (Oct 2017). No clustering of the data occurred, and all trimmed and quality filtered reads were used in the taxonomic identification step. Due to the degraded nature of the DNA from the fecal sacs we filtered the BLAST output to exclude hits with less than 95% identity across at least 100 bp of the query sequence. Data were further filtered to exclude sequences with fewer than 100 reads to eliminate degraded sequences.

To support taxonomic identifications, the discriminatory ability of the amplified gene regions was assessed. A barcode gap analysis was used to ascertain if the gene region, using publicly available data, was able to reliably place a sequence to a taxonomic group [32]. To have a barcode gap there must be less variation within a group then there is between the members of that group and all other members outside the group [33]. To provide the most robust analysis of this gap, distances between all elements were calculated and the highest difference between members within the target group was compared to the smallest distance between members of the target group and all other members outside the target group. If no gap was obtained from gap analyses, COI-5P gene region BLAST identifications were pulled back to genus. With the rbcLa gene region, taxa with no barcode gap using species level taxonomic identifications were tested between genera. If there was no apparent gap at the level of genus taxonomic identifications were pulled back to Family. Placing a higher-level taxonomic identification at genus for animals using a segment of the COI-5P and family for plants using a segment of the rbcLa follows established methods [25].

To assess if a barcode gap exists for BLAST identifications, higher taxonomic levels (family for plants and either family or genus for animals depending on the manageable size of the data set and the number of BLAST identified species from the same genus or family) were used to obtain sequence data sets from the BOLD system (manually downloaded July and August 2018 and unique identifiers are included in the S1 File). Sequences were aligned (MAFFT: 31) and trimmed to target region using MEGA (primers included in S2 File: [34]). Sequences were removed from further analysis if they had greater than 2% unknown nucleotides or if they had greater than 12 gap characters ('-') at either the 3' or 5' ends of the sequences. A distance matrix (the proportion or the number of sites that differ between each pair of aligned sequences) was constructed with the R package Ape (Ver. 4.1) dist.dna() matrix function [35]. These matrices were then used to obtain the maximum within species genetic variation and the minimum genetic distance between species of the same genus and this was completed through a custom R script (see S3 File). All within and between taxa values used to assess the specificity of the taxonomic assignments are reported (S4 File).

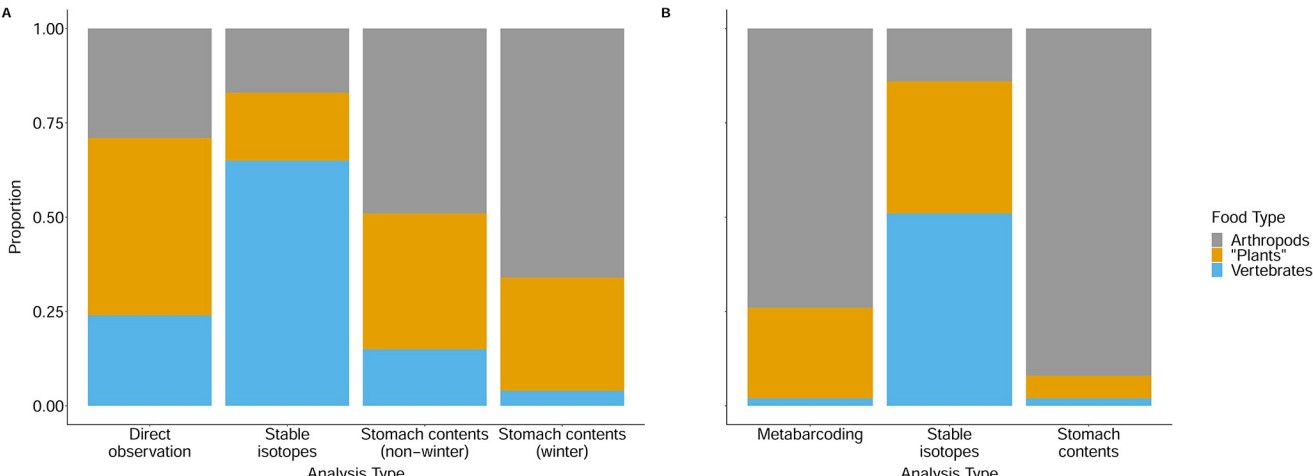

**Fig 1. Proportion of diets composed of arthropods and other invertebrates (grey), 'plants' (orange, includes plants, fungi, and molds), and vertebrate tissue (blue).** The table is divided into two sections; free flying individuals (adults and juveniles; A) and nestlings (B). We further divide our observations according to the method used to characterize diet composition (direct observation, stable isotope, and stomach contents) and present the estimated proportion of each of the three food groups in the overall diet. Additional information on identified diet items from each food type are available in S1 Table.

## Compilation of a comprehensive adult and nestling diet dataset

To facilitate a more thorough analysis and comparison of adult and nestling Canada jay diets, we combined the nestling food items identified by metabarcoding outlined above, our own and published direct observations of items taken by foraging jays across their continental range, and all items visually identified by us or others in stomach contents of both nestlings and adults, (including those identified in the 1980s by experts of the then Biosystematics Research Institute [BRI] of Canada's federal Department of Agriculture in Ottawa) into a single dataset (S1 Table). The BRI identifications (summarized in [22] but otherwise never published) included items from stomachs and regurgitated pellets of 10 APP nestlings and from 18 stomachs of adults inadvertently killed in traps set for furbearers in northern Ontario. The entire dataset included 662 items and was partitioned into three "food groups": (1) arthropods, (2) "plants"—including vascular plants, fungi, and slime moulds, and (3) vertebrate flesh—including from carcasses or small mammals and nestling birds killed by jays themselves. These items were further partitioned according to life stage (nestling *vs* "adult"), observation method (direct observation *vs* stomach contents *vs* barcoding/metabarcoding), and time of year ("winter"–Nov. 1 to Mar. 31 *vs* "non-winter"–May 1 to Oct. 31). We supplemented these data with findings from two previously published papers: a stable-isotope analysis of adult and nestling diet [10] and an observational study of adults in Alaska [36] before numerically and graphically summarizing the results (Fig 1) and using them to draw inferences about the similarities and differences between adult and nestling diets.

## Determination of cached vs "fresh" origins of food items from nestling and adult winter diets

Based on natural history considerations, we sought to determine the extent to which food items identified in nestling fecal sacs through dDNA metabarcoding or stomach contents or in the stomach contents of adults sampled in winter (Nov. 1 –Mar. 31) were likely cached or likely "fresh" (i.e., non-cached). To do this, we assigned all food items to one of four categories:

"unknown" (when nothing was known about the source organism's natural history, when we were confident the item had been misidentified, or when it had not been identified to species); "either possible" (when the food items were present and accessible year round); "likely fresh" (if the food items were widely available as non-cached items at the time when they were detected in fecal sacs or stomach contents and especially if they were unavailable in the preceding summer/fall food storage season); or "likely cached". We deemed *a priori* that the "likely cached" designation would be justified if the food item satisfied one or more of the following four criteria: (1) The food item was from a migratory species known to be entirely absent from the location where, and on the date when, the food observation was made (e.g., finding Monarch butterfly (*Danaus plexippus*) DNA in a nestling fecal sac in April), (2) The identified taxon was in a life stage known not to be present on the date of the food observation (e.g., the adult form of an insect that overwinters as an egg or larva), (3) All individuals of the source taxon are known to be underground when and where the food observation was made (e.g., a hibernating jumping mouse, Zapodidae), and (4) Snow depth on the date of the food observation would have precluded access to the item if it were still in its source location (e.g., a cranberry, *Vaccinium* sp., still on the parent plant, a few cm from the substrate). To inform our application of criterion 4, we obtained historical snow depth records from Environment and Climate Change Canada weather stations located close to the sampling location (https://climate.weather.gc.ca/historical_data/search_historic_data_e.html) and, whenever possible, supplemented them with more detailed records from the then Ontario Ministry of Natural Resources.

The final determinations of "likely cached vs likely fresh vs either possible" status was made by D.F.B. and J.H.S., using their respective botanical and entomological expertise and their specific familiarity with the natural history and phenology of northern Ontario (where we obtained most of our winter adult stomachs) and of APP (where we obtained all of our nestling fecal barcoding and stomach-contents results).

All work was approved by the Animal Utilization Committee at the University of Guelph. Further approval to collect nestling fecal samples was provided by the Canadian Wildlife Service and Ontario Parks.

## Results

### Summary of nestling diet items identified through DNA fecal metabarcoding

Metabarcoding results from the fledgling faecal samples were represented by 20 biological collections representing three gene regions. Of the three gene regions, raw sequencing results for the ITS2 focusing on fungal taxa had between 0 and 6,349 reads, rbcLa focusing on the plant taxa had between 5 and 852,304 reads, and COI-5P focusing on animal taxa had between 0 and 267,578 reads for a total of 3,761,841 (S5 File). After trimming and filtering there were 2,775,314 reads across 20 samples representing 1469 unique sequences, 1092 from the COI-5P, 370 from the rbcLa, and 7 ITS2 (S6 File). Unique sequence reads were collapsed into groups based on taxonomic assignment and taxonomic placement with enough data in public databases were evaluated using a DNA barcoding gap assessment (S4 File). After collapsing taxonomic assignments and assessing the taxonomic placement, 147 unique taxa were obtained as likely taxa from the nestling fecal sacs, 2% (3/147) were vertebrates (wood frog; *Lithobates sylvaticus* and a shrew; *Sorex cinereus*), 74% (109/147) were arthropods, and the remaining 24% (35/147) were plants. Overall, Araneae (30 spiders) and Lepidoptera (44 moths and butterflies) represented the majority total food items detected (Fig 1), but the most common food items were Ericales plants (heather and allies; most commonly blueberries in our study areas), which

were detected in 51% of the fecal sacs. The taxonomic identifications of all consumed prey items are provided in S1 Table.

## Summary description of consolidated dataset

Fig 1 summarizes and compares results across methodologies based on data in S1 Table, as well as two recently published studies [10, 36]. S1 Table lists 662 observations of Canada jay diet items that were either reported in the literature (excluding observations in 36), observed by us, or reported to us by others. Of these, 121 (18%) were "direct observations" (i.e., cases where one or more jays were seen consuming a single plant or animal species), 147 (22%) were species identified in our fecal barcoding analysis (see below), and 394 (60%) were items identified in stomach contents including 65 items from ten nestling stomachs. Within each of these three "observation-type" categories, we further partitioned observations into three broad food types (arthropods, plants, and vertebrates). Of the 121 direct observations, for example, 20 (17%) involved arthropods, 23 (19%) involved plant material, and 78 (64%) involved vertebrates. The direct observations were unique in that, for two of the direct observation sub-categories (plants and vertebrates), a further useful partitioning was possible. Eight of the 23 (35%) observations of "plants" being consumed by Canada jays involved fungi or slime moulds, hinting at the possibly under-appreciated importance of these taxa in Canada jay diets. Soft-bodied fungi cannot be identified in semi-digested stomach contents, nor could they be detected through dDNA metabarcoding of nestling fecal sacs in this study (because amplification of ITS2 was not successful) or through stable isotope analyses [10]. In a similar manner, our compilation of direct observations of the consumption of vertebrates (n = 75; S1 Table) showed that 8% (n = 6) involved vertebrate eggs, 45% (n = 34) involved carrion, and 47% (n = 35) involved live prey. Most live prey were small mammals or nestling birds but recently fledged birds were also observed to be consumed, including two that were first struck and disabled in flight [37]. Overall, nestling diet, as revealed by the combined results of fecal dDNA metabarcoding and stomach-contents analyses, had a greater proportion of arthropods than winter-adult diet (80% vs 49%), and a correspondingly lower proportion of "plant" items (18% vs 36%) and vertebrate items (2% vs 15%).

## Cached vs. fresh food determinations for diet items

Fig 2 and S2 Table summarize the 'cached-versus-fresh' results obtained from metabarcoding of nestling fecal sacs and stomach contents from adults and nestlings. Of 194 winter adult food items and 212 nestling diet items, 45% (n = 87) and 59% (n = 125), respectively, were identified to species with sufficiently well-known natural histories that we were able to further categorize the items as "likely cached", "likely fresh", or "either possible". In the subset of winter-adult food items (n = 87), we judged that 39% were "likely cached", 55% could have been cached or fresh, and only 6% were "likely fresh". In the corresponding subset of nestling food items (n = 125), we found a different pattern: 28% were "likely cached", 34% could have been either cached or fresh, and 38% were "likely fresh". "Likely-cached" food was most likely to be plant items (e.g. berries) for both winter-adult diets (57%; 32/56 total items) and nestling diets (67%; 22/33 total items). In contrast, vertebrate tissue items were the least likely to be "likely-cached" but this may reasonably be attributed to the fact that all the vertebrate items we found in winter stomachs were from species, particularly small mammals, that are active year-round, precluding us from assigning them as either cached or fresh food. Additionally, despite the relatively small proportion of "likely cached" food items in the nestling diet, it is noteworthy that, 66% (10/15) of nestling fecal sacs contained at least one "likely cached" food item, providing further support for the importance of cached food in the nestling diet of Canada jays.

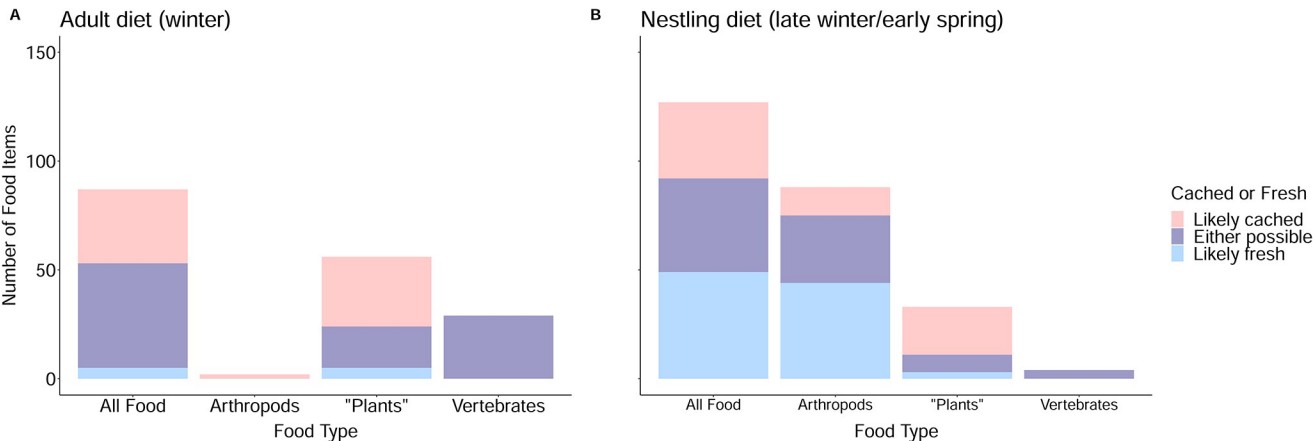

**Fig 2.** Characterization of food items as "likely cached" (light red), "either possible" (light purple), or "likely fresh" (light blue) for (A) adult Canada jay winter diet (n = 87 total observations) and (B) nestling diets using direct observations, stomach contents, and metabarcoding (n = 125 total observations). Allocation to the three categories was based on life history characteristics of food items and environmental conditions at time of observation or sample collection. The total number of food items in each of the three "cached" vs "fresh" categories are displayed under "All Food" and designations for the three major food groups (arthropods, plants, and vertebrate tissue) individually are also displayed.

## Discussion

### The value of dietary DNA metabarcoding to investigate animal diets

We used DNA metabarcoding to significantly improve our knowledge of the nestling diet of Canada jays and to strengthen evidence that stored food is important in that diet. This is a relatively new method for analyzing diets and a careful assessment of its strengths and weaknesses may, therefore, be useful (Table 1; [38]). First, this method does have limitations because it requires DNA reference libraries of potential food items [39, 40]. Incomplete libraries can lead to missed diversity as the sample reads cannot be matched to known references. In addition, samples may be missing read data from biota of interest when PCR primers are not optimized. One possible solution to primers poorly amplifying taxa, or alternatively, primers preferentially amplifying taxa, is to take a PCR-free approach and, instead, obtain sequences for all available DNA in the samples. This is sometimes referred to as a shotgun metagenomics sequencing and while this approach may help to address some primer concerns and could offer potential for prey biomass estimates, it is not without challenges in application, including high costs and the generation of large datasets leading to computationally intensive analyses [41, 42]. While increasing the amount of nucleotide sequence data obtained will provide more information, they are of little use if there are insufficient records in sequence libraries to identify unknown sequences. This lack of nucleotide sequence data was apparent in this work where some taxa were not able to be evaluated for the presence of a DNA barcode gap due to poorly populated libraries. For example, while the ITS2 region was amplified and sequenced for use in fungal identification (although our lack of success was most likely due to the degree of degradation occurring for these soft bodied organisms), the ITS region has been successfully utilized for identification of other taxa including Canadian flora [43]. However, due to the time and cost constraints to amplify the gene region using additional primers, in addition to the poorly populated sequence libraries for this taxonomic group, this approach was not feasible for this study.

Second, in combination with accurate natural-history and environmental (e.g., snow-depth) information, dDNA metabarcoding largely duplicates the ability of stomach contents analysis to plausibly assess whether or not a given item was "likely cached". Moreover, at least

**Table 1. Pros and cons of different diet analysis techniques compared in our analysis.** Our table outlines potential benefits and drawbacks of using each diet analysis technique we compared in our manuscript. Each diet analysis technique has clear advantages and potential disadvantages that should dictate when a specific approach would be most beneficial to use and when it should be potentially avoided. Study design and the purpose of the study should be considered carefully before a particular approach is used.

| Can the method. . . | direct observations | | stomach contents | fecal barcoding | stable isotopes |
|---|---|---|---|---|---|
| | (adults) | (nestlings) | (adults/nestlings) | (adults/nestlings) | (adults/nestlings) |
| . . .reliably identify all three main food groups (arthropods, "plants", vertebrates)? | Yes | No | No | Yes | Yes |
| . . .estimate the relative contributions of the main food groups to total diet? | No | No | No | No | Yes |
| . . .identify arthropod taxa? | Maybe[1] | No | Yes | Yes | No |
| . . .identify vascular plant taxa? | Yes | No | Yes | Yes | No |
| . . .identify vertebrate taxa? | Yes | No | Maybe[2] | Yes | No |
| . . .identify fungi, slime moulds (fourth food group)? | Yes | No | No | Maybe[3] | Maybe[4] |
| . . .be minimally invasive? | Yes | Yes | No[5] | Yes | Yes |
| . . .be easily used to collect observations/samples? | No | No | No | Yes | Yes |
| . . .use date and food item identity to potentially assess if it was likely cached? | No | No | Yes | Yes | No |
| . . .use date and life history stage of a food item to potentially assess if it was likely cached? | No | No | Yes | No | No |

[1]in most cases it is difficult to identify arthropods being consumed through direct observations, but certain taxa may be possible to identify

[2]vertebrate taxa can be identified if stomach contents contain bones or teeth that may facilitate identification

[3]advances in metabarcoding techniques now mean that identification of fungi and slime moulds should be possible in barcoding studies, although the approaches employed in this study did not allow us to identify fungi in the Canada jay diet

[4]Freeman et al. 2021 could not distinguish slime moulds or fungi from other groups due to the isotopes used, but their identification may be possible with additional isotopes

[5]not invasive if stomach contents can be collected from dead animals or in the form of stomach castings (e.g. Fig 3f)

in principle, stomach contents analysis would outperform barcoding if it found exoskeletal fragments corresponding to an arthropod's life-history stage that was unavailable in nature on the date when the fragments were found; dDNA metabarcoding cannot aspire to such determinations because it can only provide species identity, not its life-history stage (Table 1). As a practical matter, however, we found no examples where this potential handicap detracted from dDNA metabarcoding's otherwise superior ability to identify taxa ingested by Canada jays and to determine which items had been "likely cached". Not only does metabarcoding permit the identification of soft-tissue or small food items that are easily missed by stomach contents analysis [44–46], but also, compared to stomach contents analysis, it allows more samples to be collected, far more easily, and importantly, without sacrificing individuals. In just three seasons (2015–2017) of measuring nestlings, we were able to double the number of identified nestling diet items that we had obtained during the previous 40 years through chance acquisitions of stomach contents and castings. A further advantageous feature is that the technique could be extended to more accurately assess the diets of not just nestling Canada jays, but also those of adults, including the winter diets that are of greatest interest. We have no doubt that it would be eminently feasible to capture winter adults and safely hold them in a suitably appointed small cage until they produced a fecal sample. By the same reasoning, it should be possible, in principle, to use fecal dDNA metabarcoding far beyond the limited use we have made of it here and we encourage others to do so.

## Characterization of the Canada jay diet by season and life history stage

The four methods we used to elucidate diet (direct observations, stomach contents analysis, stable isotope analysis, and fecal metabarcoding; Fig 1) all indicated that Canada jays consume

arthropods, "plants" (almost entirely berries), and vertebrates, but each method provided different estimates for the overall portion of each food type in the diet. For example, it is likely that the high proportions of vertebrate flesh suggested by our compilation of "direct observations" (65%) and stable isotope analysis of food-dependent APP juveniles (72%; 10) are both overestimates. Stable isotope analyses relied on a fractionation factor derived from an unrelated species (Yellow-rumped warbler; *Setophaga coronata*), which could have resulted in a mischaracterization of diet groups and their contributions to Canada jay diet [10]. Similarly, our compilation of direct observations almost certainly exaggerates the importance of vertebrate flesh in the diet (and minimizes that of arthropods) because observers are more likely to report consumption of familiar vertebrate taxa than of tiny, quickly ingested invertebrates. In contrast, the much lower proportion of vertebrates (24%) reported in a direct-observation study in Denali, Alaska [36] is probably more accurate because it is based on a much larger sample and because the observers made concerted efforts to record all food acquisitions by focal individuals. However, we also note that differences in diet composition between the APP and Denali population could simply be due to differences in prey composition or climate. Despite potential bias associated with direct observations, one major advantage of this approach is that it is able to register the consumption of fleshy fungi (Fig 3a) and slime moulds (Fig 3b), distinguish between incidents of scavenging versus actual predation (Fig 3c and 3d), and record unusual foraging methods (e.g. flycatching or wading into shallow water to capture tadpoles [47] and larval salamanders [48]).

The only presently available data permitting a comparison of winter and "non-winter" (May 1 –Oct. 31) Canada jay diets are from stomach contents and stomach castings (Fig 3e). Given the frigid realities of boreal forest winters, it is noteworthy that arthropod and plant items (Fig 3f) accounted for 49% and 36%, respectively of items identified in winter-adult stomachs, compared to corresponding figures of 66% and 30% in non-winter stomachs.

In the Canada jay nestling period, arthropods accounted for even greater proportions of the items identified in nestling stomachs and feces (92% and 74%, respectively) compared to the winter diet of adults, possibly reflecting a particular preference in parents for protein-rich arthropods for their rapidly growing nestlings, as has been found in many other bird species [49, 50]. Caution is warranted, however, before concluding that arthropods are overwhelmingly important in the diet of Canada jay nestlings. Notably, dDNA metabarcoding cannot rule out the possibility of secondary predation [51]. Also, in their stable-isotope analysis of nestling tissues, Freeman et al. [10] found that only 14% of the nestling diet was attributable to invertebrates whereas 51% of the food given to nestlings was believed to be of vertebrate or human-food origin. Notwithstanding methodological issues with stable isotope analysis noted above, this may well be the more meaningful representation of the nestling diet because stable isotopes give an estimate of volume of each food class whereas the analysis of nestling stomach contents or feces yields numbers of separate, identifiable food items or taxa contained in each stomach or fecal sac without an indication of the proportion of each food item. Consider that, if a Canada jay consumed three individuals of each of five species of arthropods and an amount of flesh and identifiable small bones from a single mouse species with an equal nutritional value and mass to that of the arthropods, stomach-contents analysis would potentially indicate, correctly, that the jay had consumed 15 arthropod items and one vertebrate item, dDNA metabarcoding would indicate that the diet included 5 arthropod taxa and one vertebrate, and stable-isotope analysis would indicate, also correctly, that arthropods and vertebrates contributed equally to the jay's nutrition. Despite these persisting uncertainties in how best to characterize Canada jay diets, we believe our results make clear that nestling and adult diets are at least roughly similar and that, even in winter, Canada jays somehow have access to

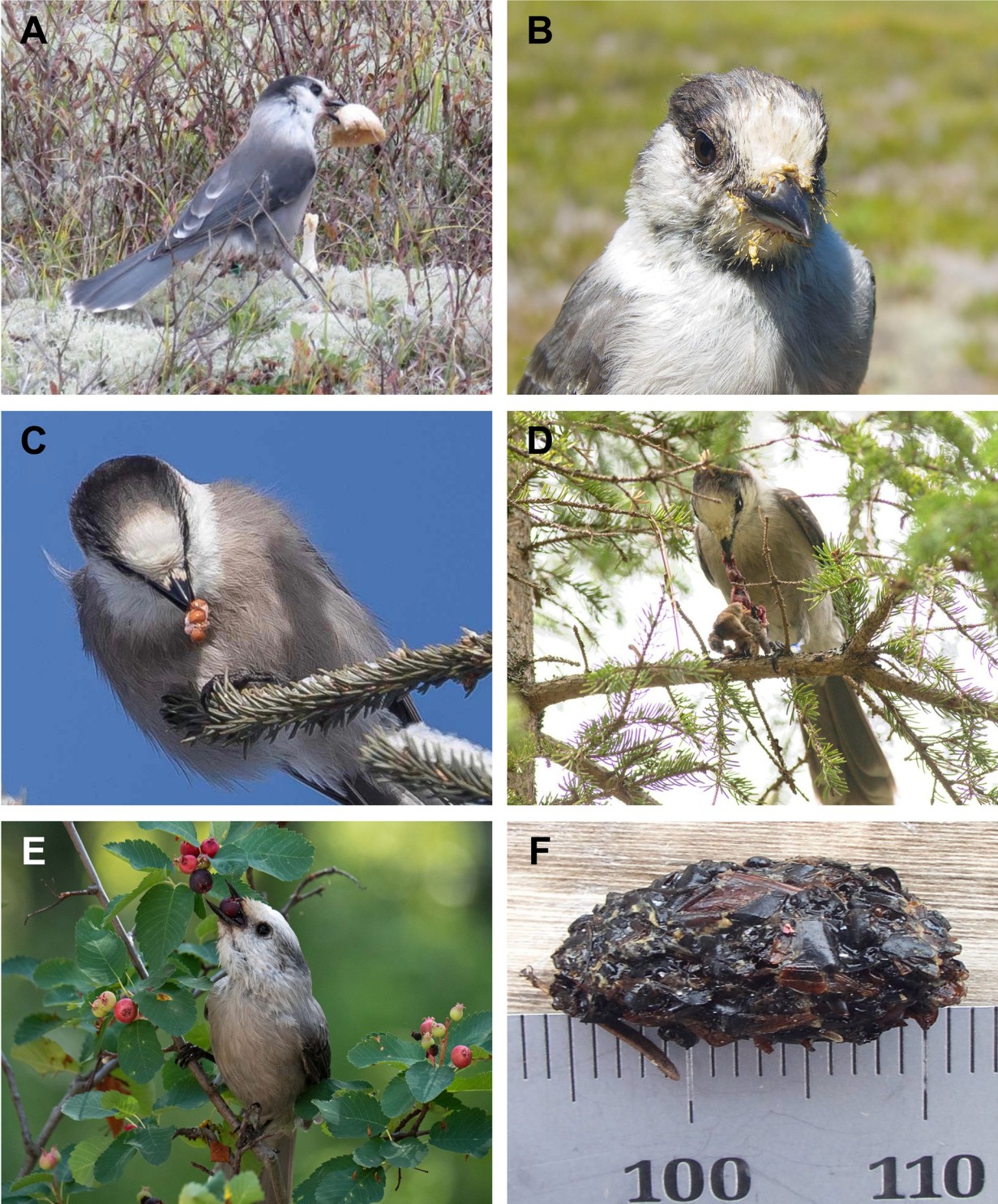

**Fig 3. Direct observations made of Canada jays foraging on a wide range of food items.** *A)* Canada jay (boreal morphotype) in Algonquin Provincial Park, Ontario feeding on *Amanita muscaria* (Photo by Langis Sirois, November 5, 2018), *B)* Canada jay (Pacific morphotype) with yellow residue around bill from recent consumption of the slime mould (*Fuligo septica*) on Vancouver Island, British Columbia (Photo by Dan Strickland, August 1, 2020), *C)* Canada jay (boreal morphotype) in Algonquin Provincial Park, Ontario regurgitating Choke Cherry (*Prunus virginiana*) seeds on a date (January 20, 2020) when cherries would only be available as cached items (Photo by Michael Runtz), *D)* Canada jay in Algonquin Provincial Park, Ontario dismembering a recently caught shrew (*Sorex sp*.) (Photo by Ann Brokelman, August 27, 2016), *E)* Canada jay

(Rocky Mountain morphotype) in Grand Teton National Park, Wyoming consuming a Saskatoon berry (*Amelanchier alnifolia; fide* DFB; Photo by Susan Elliot and used with permission from the Macaulay Library at the Cornell Lab of Ornithology (ML478606261), August 26, 2022). *F)* Canada jay stomach casting apparently consisting of arthropod exoskeletal fragments (Strathcona Provincial Park, Vancouver Island, British Columbia; Photo by Dan Strickland, June 26, 2021).

berries and arthropods that would seem to be primarily, if not exclusively, available only in the snow-free part of the year.

## Contribution of stored food to adult-winter and nestling diets

By combining the identification to species level of many items found in the stomachs and feces of winter adults and nestlings with natural history information about those food species, we deemed that 39% of winter-adult items had likely been cached and only 6% were likely found as "fresh items". By contrast, we deemed that only 28% of items detected in nestling stomachs and feces had likely been cached and at least 38% had been consumed as fresh items. We attribute the much greater proportion of apparently fresh items in the nestling diet to the fact that Canada jay nestling period in our study area typically straddles the disappearance of snow cover in mid- to late-April and, in our case, half of our nestling stomachs and fecal sacs were collected considerably later than that, even as late as May 19 (from a replacement nest). Derbyshire et al. [20] previously pointed out that once snow cover has disappeared, Canada jay parents forage for their nestlings on the forest floor where food storage has never been observed and Swift et al. [36] similarly reported that Canada jays in Denali, Alaska "responded to a record-setting warm spring by directing their foraging efforts away from cache recovery and towards the emergence of fresh food". Together these results lend further support to the suggestion by Strickland and Ouellet [22] that food storage is the key behaviour in Canada jays permitting not only their high winter survival and territorial fidelity, but also their high nesting success (thanks to the use of stored food as critical "emergency food" in late springs or during the not unusual snow and ice storms that occur in the late-winter Canada jay nesting season).

Our work demonstrates the value of combining methods (e.g., direct observation, stomach contents, and DNA metabarcoding) when assessing the composition of the Canada Jay's nestling and adult diets. We believe a similar approach could be applied to other free-living species to advance our knowledge of the diet of wild birds. Such knowledge is integral to understanding how individuals interact with their environments and how diet, and subsequently individual performance, may change across habitats and with climate change [52].

## Supporting information

**S1 Table. Summary of all food items known to be consumed by Canada jays.** The date and location the observation was made is included for each observation in addition to the identification of the food item being consumed. We also summarize additional information related to who made the record, what method was used to make the observation ("DO"–direct observation, "SC"–stomach content analysis, "FS"–metabarcoding of fecal samples), the age of the individual which consumed a food item (e.g., adult, juvenile, or nestling), and which of our four diet groups the observation belongs in ("A"–arthropod, "P"–"plants", "V"–vertebrate tissue). Finally, for observations made during the winter and spring, we describe whether a food item was "likely-cached", "likely fresh", or if it could be either cached or fresh. Observations labeled with "ROM" represent stomach samples from the Royal Ontario Museum collection. (DOCX)

**S2 Table. Summary table of diet observations made during the winter for both adult and nestling Canada jays.** Diet items were designated as "likely cached", "likely fresh", or "either possible" based on natural history information available for each food item. We further classified diet items based on whether they were consumed by adults or nestlings, and based on the method used to collect the diet item.
(XLSX)

**S1 File. Barcode gap records.** Custom dataset containing sequence data sets from the BOLD system and used to assess if a barcode. Gap exists for BLAST identifications.
(TSV)

**S2 File. A complete list of primers used during PCR amplification.** All primers used in our analysis are outlined here in addition to being described in the method section. Available at 10.6084/m9.figshare.25422847.
(TXT)

**S3 File. R script used to calculate genetic distances between species identified using DNA metabarcoding.** Available at 10.6084/m9.figshare.25422847.
(DOCX)

**S4 File. Within and between taxa values used to assess the specificity of taxonomic assignments from our DNA metabarcoding analysis.**
(TSV)

**S5 File. Raw, unfiltered sequencing results from our DNA metabarcoding analysis.**
(TSV)

**S6 File. Trimmed and filtered BLAST results.**
(TSV)

## Acknowledgments

We would like to thank numerous field assistants who collected diet information and demographic data from Canada jays in Algonquin Provincial Park. We would also like to thank the Ontario Ministry of Northern Development, Mines, Natural Resources and Forestry for providing snow depth data. Finally, thank you to Langis Sirois, Michael Runtz, Ann Brokelman, the Macaulay Library, and Susan Elliot for allowing us to use their photos.

## Author Contributions

**Conceptualization:** Alex O. Sutton, Dan Strickland, Nikole E. Freeman, D. Ryan Norris.

**Data curation:** Robert G. Young, Robert Hanner.

**Formal analysis:** Alex O. Sutton, Dan Strickland, Jacob Lachapelle, Robert G. Young, Robert Hanner, Daniel F. Brunton, Jeffrey H. Skevington.

**Methodology:** Alex O. Sutton, Dan Strickland, Jacob Lachapelle, Robert G. Young, Robert Hanner, Daniel F. Brunton, Jeffrey H. Skevington, D. Ryan Norris.

**Supervision:** Alex O. Sutton, D. Ryan Norris.

**Visualization:** Alex O. Sutton, Nikole E. Freeman.

**Writing – original draft:** Alex O. Sutton, Dan Strickland, Jacob Lachapelle, Nikole E. Freeman, D. Ryan Norris.

**Writing – review & editing:** Alex O. Sutton, Dan Strickland, Jacob Lachapelle, Robert G. Young, Robert Hanner, Daniel F. Brunton, Jeffrey H. Skevington, Nikole E. Freeman, D. Ryan Norris.

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
