## [Decision Letter · Decision Letter 0]

17 Sep 2023

PONE-D-23-24828Fecal DNA metabarcoding helps characterize Canada jay diet and confirms the reliance of stored food for winter survival and breedingPLOS ONE

Dear Dr. Sutton,

Thank you for submitting your manuscript to PLOS ONE. After careful consideration, we feel that it has merit but does not fully meet PLOS ONE’s publication criteria as it currently stands. Therefore, we invite you to submit a revised version of the manuscript that addresses the points raised during the review process.

We look forward to receiving your revised manuscript.

Kind regards,

Petr Heneberg

Academic Editor

PLOS ONE

Journal Requirements:

"We would also like to thank NSERC (DRN, NEF), Ontario Parks (AOS, DRN, NEF), Friends of Algonquin Park, Wildlife Conservation Society Canada (AOS, NEF), and the Ministry of Natural Resources and Forestry (AOS, DRN, NEF) for providing funding and logistical support for our project. "

3. Please expand the acronym “NSERC” (as indicated in your financial disclosure) so that it states the name of your funders in full.

"We would like to thank countless field assistants who collected diet information and demographic data from Canada jays in Algonquin Provincial Park. We would also like to thank NSERC, Ontario Parks, Friends of Algonquin Park, Wildlife Conservation Society Canada, and the Ministry of Natural Resources and Forestry for providing funding and logistical support for our project. Finally, we would like to thank the Ontario Ministry of Northern Development, Mines, Natural Resources and Forestry for providing snow depth data."

"We would also like to thank NSERC (DRN, NEF), Ontario Parks (AOS, DRN, NEF), Friends of Algonquin Park, Wildlife Conservation Society Canada (AOS, NEF), and the Ministry of Natural Resources and Forestry (AOS, DRN, NEF) for providing funding and logistical support for our project. "

5. We note that you have referenced (unpublished) on page 2, which has currently not yet been accepted for publication. Please remove this from your References and amend this to state in the body of your manuscript: (ie “Bewick et al. [Unpublished]”) as detailed online in our guide for authors

Reviewers' comments:

Reviewer's Responses to Questions

**Comments to the Author**

1. Is the manuscript technically sound, and do the data support the conclusions?

Reviewer #1: Yes

Reviewer #2: Yes

Reviewer #3: Partly

Reviewer #4: Partly

Reviewer #5: Yes

2. Has the statistical analysis been performed appropriately and rigorously? 

Reviewer #1: Yes

Reviewer #2: No

Reviewer #3: No

Reviewer #4: I Don't Know

Reviewer #5: N/A

3. Have the authors made all data underlying the findings in their manuscript fully available?

Reviewer #1: Yes

Reviewer #2: Yes

Reviewer #3: No

Reviewer #4: No

Reviewer #5: Yes

4. Is the manuscript presented in an intelligible fashion and written in standard English?

Reviewer #1: Yes

Reviewer #2: Yes

Reviewer #3: Yes

Reviewer #4: Yes

Reviewer #5: Yes

5. Review Comments to the Author

Reviewer #1: This is a well-written paper. It is particularly valuable in that it compares the results of 4 methods that have been used widely. I supports the view that a substantial portion of the jay's diet come from cached food.

Reviewer #2: The presented manuscript describes original research on determining the diet of Canada jay. As the identification of wild animal’s diet is notoriously difficult, the authors venture into new methodology, namely metabarcoding, to help inform their inferences and determine how important food caches are to the Canada jay’s winter survival. They combine this DNA-based method with more traditionally used stomach contentanalyses, stable isotopes and direct observations.

Article is written in a clear and concise way, with good structure - reasoning is fairly transparent and followed throughout the text. Methods and analysis are only partially well described (see comments below). Results are presented with clarity and the overall impression of the manuscript is positive; however, it could definitely benefit from additional analyses/visualizations.

Major issues

There is the need to describe metabarcoding approach more. The authors bravely dive into this methodology, new to the field. Even more, they encourage others to use metabarcoding to widen the scope of their studies. It is very commendable, however, comes with large responsibility. The authors are basically setting the standards for the whole field. They have to present the method comprehensively and understandably. At the moment, the manuscript elaborates on the selection of markers (important matter) but lack entirely any description of DNA extraction, amplification and sequencing (ironically DNA extraction, amplification and sequencing is the deadline of the paragraph; line 130). The authors refer to Posser and Hebert 2017 for all details. In my opinion, which is not alienated, the reader deserves to have a short description of what happened with the samples, as it is necessary to understand the results.

The same is true for the Bioinformatics section of the manuscript (starting at line 153). Information about filtering to a minimum length of x is not informative because we did not learn what how the samples were sequenced. We don’t know what technology was used (Illumina?) and what length of reads was generated?? We’re completely uniformed in this respect. Bioinformatic processing is crucial in metabarcoding projects and has to be reproducible. It is necessary that to publish your scripts/pipelines in some form. Or at least describe it in detail. In line 157 you mention custom reference databases. Custom in what way? Please give readers some details on how you made those databases, so that they can reproduce your analysis.

Two main figures of the text, Fig 1 and 2, are showing a similar type of data but Fig 1 in percentages and Fig 2 in number of food items. It left me a bit perplexed, as I did not understand the reason behind this choice. Relative abundance of anything (like food types Fig. 1) can be generally misleading and I thought it would be more straight-forward to use just absolute numbers in both figures. But perhaps I’m just unfamiliar with the type of analyses you’re doing and I am definitely open to read your reasoning.

Lastly, you make a very good point in the discussion about advantages, and shortcomings of different methods and basically you give great insights on how use those very different sources of data, combine and make sense out it all. I think your paper would profit immensely from a figure summarizing some of the aspects of it. What methods exaggerate importance of some food sources and diminish others? And it could be the place to highlight how metabarcoding improves the overall picture. Obviously, this is just a suggestion but I think it will make your work much more impactful.

Minor issues

The work compiles a large dataset of direct observations of what birds were consuming. It is a very impressive work done with great detail. In fact, I think the dataset is really worth sharing with other researches and should be formatted in a way that is easily accessible. Right now is a Word .doc submitted as a supplementary information. I think it would make much more sense to share it in a .CSV or .TSV format, perhaps GitHub is a good place to upload it? Additionally, the authors could use GitHub to share their bioinformatic pipeline as well.

Line 127: samples are stored in a freezer and then in another freezer. What is the temperature of the first freezer?

Line 184: A simple distance matrix. What distance? Euclidean? Jaccard? B-C? Specify please.

Line 185: Provide version of the package and citation.

Line 277: metabarcoding, not just barcoding.

Line 301: when reference libraries are incomplete you cannot identify some species down to species level but it doesn’t cause misidentification automatically.

Line 302: PCR primers have what is called “primer bias” and some taxa might be missed completely by certain primers, while other taxa will be amplifying strong. It is recommended to use wide-spectrum primers; not optimized for the taxa of interest. I think I understand what you meant in this sentence but the wording suggests something different. Consider clarifying.

Reviewer #3: The authors can find more detailed comments in the manuscript attached. I do feel that the following points needs to be addressed to make this work meaningful and add scientific value to the study:

1. The first concern is with the description of the metabarcoding methods that were used. The description of the methods are incomplete (PCR protocol, sequencing done). The description of the bioinformatic methods used are incomplete (software packages). Which model was chosen to create the distance matrices? There is a reference to supplemental data (distance matrices) that were not available. The methods used to assign taxonomy are not described clearly.

2. It is noted that custom databases were created from data obtained from BOLD and Genbank. However, there are no reference species lists noted that were used to create these custom reference databases. Furthermore the accession numbers of the sequences used to create the reference bases are missing as well as the criteria used to choose the sequences for these custom reference databases.

3.In the results section the metabarcoding results are again incomplete. Of the 147 taxa detected give a list of how many and which taxa, up to family level, genus level and species level? What was the results of the gap analyses for each of the custom databases?

4.In the discussion the authors refer to only the detected food items and not to the relative abundances of the food items that can shed much more light on the diets especially during winter and non-winter and adults vs nestlings. It can even shed more light on cached vs fresh food items. Through the use of FOO and RRA biases could be explored as well as the possibility of using cached items.

5.In the Abstract no reference is made to the type of food items (plants, invertebrates) or how many taxa were detected. There is no mention of dmetabarcoding methods which forms a critical part of the study.

Reviewer #4: I would like to thank the authors and the editor to give me the opportunity to read the manuscript titled “Fecal DNA metabarcoding helps characterize Canada jay diet and confirms the reliance of stored food for winter survival and breeding”.

I found the manuscript very interesting and I think the study has merit, especially in combining molecular and non-molecular techniques.

My expertise is mostly revolving around DNA metabarcoding, so I focused my review on this technique.

Unfortunately, I am afraid the manuscript in its current form requires a major revision since most of the information on the use of metabarcoding are missing or are incomplete.

However, I strongly encourage the authors to address my comments and concerns and provide a revised version of their work, since it will be an important tool for future research.

Main comments:

If the ITS marker could not be amplified (despite following published protocols), the authors should probably remove any reference to it everywhere in this manuscript.

Metabarcoding methods needs to be improved:

Page 7: No information is provided for the sequencing. I think it is very important to provide this kind of information in detail:

- What did the authors do after PCR amplification?

- How did they prepare the sequencing library?

- Were the samples run in duplicate?

- Were there controls to account for plant/fungi present in the nest but not part of the diet?

- Were dual unique indexes used?

- Who performed the sequencing (in-house or outsourced)?

- What platform did the authors used to perform the sequencing?

- Were all markers pooled in the same sequencing run?

Page 8: Metabarcoding reads quality control is not really mentioned, except for size and primer trimming. Were there any chimera removal? Were coding genes checked for stop codons? Especially for fungi, was there any threshold-based OTU clustering?

Page 9: please include software references for R/RStudio and package version for each of the packages used.

Results:

The authors should summarise the fate of the raw reads obtained using metabarcoding, ideally for each genetic marker. For example:

“A total of XX millions raw reads was generated from XX samples. Of these, X thousand reads passed quality controls. A total of XX taxa could be identified based on COI, XXX taxa for ITS, XXX taxa for Rbcl”

Raw data for metabarcoding analysis should be made publicly available.

Minor comments:

Line 132: Change “molecular regions” to “gene regions”.

Line 133: Italicise the name of the genes when reported in full. “Cytochrome c Oxidase” should be “Cytochrome c Oxidase” and so on.

Line 139: the expression “molecular gene region” is used a number of times. I think the use of “molecular” in this case is redundant since any gene region is “molecular” in a way. Can you please remove all instances of “molecular” and leave only “gene region”? At line 138, “molecular marker” is correct and can be used since it specify the type of marker.

Lines 144-145: repetition of “well established” twice over two sentences. If you could remove one instance, it may make the sentence flow better.

Line 147: Change “molecular regions” to “gene regions”.

Line 155: This sentence is missing something “Primers were trimmed from each read and filtered again by a minimum size of 100 bp”. The authors trimmed the primers and then filtered.. the reads? As it stands, the subject of the second sentence is still “primers”.

Line 244 (and line 226): Following the rules of the International Code of Zoological Nomenclature, only names of genera and species should be italicised, but not families and orders. Araneae and Lepidoptera (and Zapodidae) should not be italicised. Also the abbreviation “sp.” should not be italicised (line 227).

Line 246: I am less familiar with plant taxonomy, but I think the above rule also applies there. Ericales should not be italicised.

Line 247: no need of upper case for “blueberries”.

Reviewer #5: This is a very interesting work, and only a few issues need the author's attention.

1. More literature on DNA fecal metabarcoding needs to be added in the introduction section.

2. The methods section needs to be greatly simplified, especially the section on sequencing and bioinformatics. In addition, it is necessary to indicate which sequencing platform was used for sequencing.

3. What does "nestlings" mean？How did this research get samples of nestlings？Do young birds at different stages of development need to be distinguished?

4. Why not use ITS2 to amplify plants, as suggested in this paper “Validation of the ITS2 region as a novel DNA barcode for identifying medicinal plant species”?

5. Could you consider using "shotgun metabarcoding" for further research? At least the relevant discussion should be added to the discussion section. Relevant literature can be used for reference, for example “The species identification in traditional herbal patent medicine, Wuhu San, based on shotgun metabarcoding”.

6. PLOS authors have the option to publish the peer review history of their article (what does this mean?). If published, this will include your full peer review and any attached files.

Reviewer #1: No

Reviewer #2: No

Reviewer #3: **Yes: **Sandra Barnard

Reviewer #4: No

Reviewer #5: No

NOTE: Some reviewer comments were submitted as an attachment file, they are attached to this email and accessible via the submission site. Please log into your account, locate the manuscript record, and check for the action link "View Attachments". If this link does not appear, there are no attachment files.

---

## [Author Response · Author response to Decision Letter 0]

21 Nov 2023

Response to Reviewers

We thank the Associate Editor and five reviewers for their valuable feedback and comments on the manuscript. Below, we have addressed all of the comments from reviewers. 

Reviewer #1: This is a well-written paper. It is particularly valuable in that it compares the results of 4 methods that have been used widely. I support the view that a substantial portion of the jay's diet come from cached food.

Author response: We are glad the reviewer enjoyed our paper and thank them for their time reviewing the manuscript.

Reviewer #2: The presented manuscript describes original research on determining the diet of Canada jay. As the identification of wild animal’s diet is notoriously difficult, the authors venture into new methodology, namely metabarcoding, to help inform their inferences and determine how important food caches are to the Canada jay’s winter survival. They combine this DNA-based method with more traditionally used stomach content analyses, stable isotopes and direct observations.

Article is written in a clear and concise way, with good structure - reasoning is fairly transparent and followed throughout the text. Methods and analysis are only partially well described (see comments below). Results are presented with clarity and the overall impression of the manuscript is positive; however, it could definitely benefit from additional analyses/visualizations.

Author response: We thank reviewer #2 for reviewing our manuscript and providing valuable feedback on our analysis.

Major issues

There is the need to describe metabarcoding approach more. The authors bravely dive into this methodology, new to the field. Even more, they encourage others to use metabarcoding to widen the scope of their studies. It is very commendable, however, comes with large responsibility. The authors are basically setting the standards for the whole field. They have to present the method comprehensively and understandably. At the moment, the manuscript elaborates on the selection of markers (important matter) but lack entirely any description of DNA extraction, amplification and sequencing (ironically DNA extraction, amplification and sequencing is the deadline of the paragraph; line 130). The authors refer to Posser and Hebert 2017 for all details. In my opinion, which is not alienated, the reader deserves to have a short description of what happened with the samples, as it is necessary to understand the results.

The same is true for the Bioinformatics section of the manuscript (starting at line 153). Information about filtering to a minimum length of x is not informative because we did not learn what how the samples were sequenced. We don’t know what technology was used (Illumina?) and what length of reads was generated?? We’re completely uniformed in this respect. Bioinformatic processing is crucial in metabarcoding projects and has to be reproducible. It is necessary that to publish your scripts/pipelines in some form. Or at least describe it in detail. In line 157 you mention custom reference databases. Custom in what way? Please give readers some details on how you made those databases, so that they can reproduce your analysis.

Author response: As suggested, we have revised the methods section to provide more of an overview of the methods used to process and sequence samples used in our DNA metabarcoding analysis (lines 128 – 187). We highlight changes made throughout the methods section below in response to specific comments.

Two main figures of the text, Fig 1 and 2, are showing a similar type of data but Fig 1 in percentages and Fig 2 in number of food items. It left me a bit perplexed, as I did not understand the reason behind this choice. Relative abundance of anything (like food types Fig. 1) can be generally misleading and I thought it would be more straight-forward to use just absolute numbers in both figures. But perhaps I’m just unfamiliar with the type of analyses you’re doing and I am definitely open to read your reasoning.

Author response: We chose to include Figure 1 because stable isotope analyses of tissue can only estimate the proportion of each food type, not the absolute number of diet items. For the reasons reviewer #2 points out, in figure 2 we present the absolute numbers without the stable isotope results. 

Lastly, you make a very good point in the discussion about advantages, and shortcomings of different methods and basically you give great insights on how use those very different sources of data, combine and make sense out it all. I think your paper would profit immensely from a figure summarizing some of the aspects of it. What methods exaggerate importance of some food sources and diminish others? And it could be the place to highlight how metabarcoding improves the overall picture. Obviously, this is just a suggestion but I think it will make your work much more impactful.

Author response: As suggested, we have created a table to describe pros and cons of each analysis and have included this in the revised version of the manuscript (Table 1).

Minor issues

The work compiles a large dataset of direct observations of what birds were consuming. It is a very impressive work done with great detail. In fact, I think the dataset is really worth sharing with other researches and should be formatted in a way that is easily accessible. Right now is a Word .doc submitted as a supplementary information. I think it would make much more sense to share it in a .CSV or .TSV format, perhaps GitHub is a good place to upload it? Additionally, the authors could use GitHub to share their bioinformatic pipeline as well.

Author response: We plan on making the data from our analysis available to anyone interested in using it. Further, data generated from our DNA metabarcoding and the complete list of diet items identified will also be deposited in an online repository once the paper is accepted. The code used in the analyses are available online (via GitHub and CRAN) and are included in the text of the updated methods section.

Line 127: samples are stored in a freezer and then in another freezer. What is the temperature of the first freezer?

Author response: The movement of the samples has been clarified and the first freezer temperature has been included (line 126). 

Line 184: A simple distance matrix. What distance? Euclidean? Jaccard? B-C? Specify please.

Author response: As suggested, we have revised this sentence to include the phrase “…, the proportion or the number of sites that differ between each pair of aligned sequences,…” (lines 224 – 226).

Line 185: Provide version of the package and citation.

Author response: Thank you, we have included the Ape version number that was used in the analysis and note that the citation for the ape package is the citation at the end of the sentence. The updated text reads “matrix was constructed using with the R package Ape (Ver. 4.1) and the dist.dna() matrix function (33)” (lines 225 – 226).

Line 277: metabarcoding, not just barcoding.

Author response: As suggested, we have added metabarcoding in addition to the barcoding already mentioned (line 334).

Line 301: when reference libraries are incomplete you cannot identify some species down to species level but it doesn’t cause misidentification automatically.

Author response: We agree and “misidentifications” was perhaps not the best choice in words. We have revised this section to read: “First, this method does have limitations because it requires DNA reference libraries of potential food items (39,40). Incomplete libraries can lead to missed diversity as the experimental reads cannot be matched to known references. In addition, experimental results may be missing read data from biota of interest when PCR primers are not optimized.” (lines 357 – 361).

Line 302: PCR primers have what is called “primer bias” and some taxa might be missed completely by certain primers, while other taxa will be amplifying strong. It is recommended to use wide-spectrum primers; not optimized for the taxa of interest. I think I understand what you meant in this sentence but the wording suggests something different. Consider clarifying.

Author response: Thank you, please see above comment as we have updated the writing and clarified our point. The updated text now reads: “In addition, experimental results may be missing read data from biota of interest when PCR primers are not optimized. One possible solution to primers poorly amplifying taxa, or alternatively, primers preferentially amplifying taxa, is to take a PCR-free approach and, instead, obtain sequences for all available DNA in the samples. This is sometimes referred to as a shotgun metagenomics sequencing and while this approach may help to address some primer concerns and could offer potential for prey biomass estimates it is not without challenges in application, including high costs and the generation of large datasets leading to computationally intensive analyses (41, 42).” Lines 359 – 366). 

Reviewer #3: The authors can find more detailed comments in the manuscript attached. I do feel that the following points needs to be addressed to make this work meaningful and add scientific value to the study:

1. The first concern is with the description of the metabarcoding methods that were used. The description of the methods are incomplete (PCR protocol, sequencing done). The description of the bioinformatic methods used are incomplete (software packages). Which model was chosen to create the distance matrices? There is a reference to supplemental data (distance matrices) that were not available. The methods used to assign taxonomy are not described clearly.

Author response: As suggested, we have substantially revised the methods section based on reviewer comments to clarify the metabarcoding and bioinformatic methods used. We outline the complete PCR protocol and the sequencing methods used to conduct DNA metabarcoding (lines 128 – 187). Additionally, we provide a complete description of our bioinformatic methods, including software packages used in our analysis (Lines 189 – 230). We describe how distance matrices were created (“A simple distance, the proportion or the number of sites that differ between each pair of aligned sequences, matrix was constructed with the R package Ape (Ver. 4.1) dist.dna() matrix function (35). These matrices were then used to obtain the maximum within species genetic variation and the minimum genetic distance between species of the same genus and this was completed through a custom R script” Lines 224-228) and methods used to assign taxonomy (“To support taxonomic identifications, the discriminatory ability of the amplified gene regions was assessed. A barcode gap analysis was used to ascertain if the gene region, using publicly available data, was able to reliably place a sequence to a taxonomic group. To have a barcode gap there must be less variation within a group then there is between the members of that group and all other members outside the group (33). To provide the most robust analysis of this gap, distances between all elements were calculated and the highest difference between members within the target group is compared to the smallest distance between members of the target group and all other members outside the target group. COI gene region BLAST identifications were pulled back to genus if no gap was obtained from our gap analyses. With the rbcLa gene region, taxa with no barcode gap using species level taxonomic identifications were tested between genera. If there was no apparent gap at the level of genus taxonomic identifications were pulled back to Family. Placing a higher-level taxonomic identification at genus for animals using a segment of the COI and family for plants using a segment of the rbcLa follows established methods (25).” Lines 202-215).

2. It is noted that custom databases were created from data obtained from BOLD and Genbank. However, there are no reference species lists noted that were used to create these custom reference databases. Furthermore the accession numbers of the sequences used to create the reference bases are missing as well as the criteria used to choose the sequences for these custom reference databases.

Author response: Thank you, we agree more clarity is necessary here. We have significantly increased the content in the methods section to address this and other methods-related comments. We have noted the content of the reference databases and referenced the paper where our methods were derived and the updated text reads: “After these steps, each read was taxonomically assigned using BLAST and three databases: all flowering plant rbcLa sequences from BOLD, all insect COI sequences from GenBank, and all fungi ITS2 sequences from BOLD (Oct 2017).” lines 194 – 197).

3.In the results section the metabarcoding results are again incomplete. Of the 147 taxa detected give a list of how many and which taxa, up to family level, genus level and species level? What was the results of the gap analyses for each of the custom databases?

Author response: As suggested, we have added this information to the results section ( “Of the three gene regions, raw sequencing results for the ITS2 focusing on fungal taxa had between 0 and 6349 reads, rbcLa focusing on the plant taxa had between 5 and 852,304 reads, and COI-5P focusing on animal taxa had between 0 and 267,578 reads for a total of 3,761,841 (Supplemental file S5). After trimming and filtering there were 2,775,314 reads across 20 samples representing 1469 unique sequences, 1092 from the COI-5P, 370 from the rbcLa, and 7 ITS2 (Supplemental file S6). Unique sequence reads were collapsed into groups based on taxonomic assignment and taxonomic placement with enough data in public databases were evaluated using a DNA barcoding gap assessment (Supplemental file S4). After collapsing taxonomic assignments and assessing the taxonomic placement, 147 unique taxa were obtained as likely taxa from the nestling fecal sacs, 2% (3/147) were vertebrates (wood frog; Lithobates sylvaticus and a shrew; Sorex cinereus), 74% (109/147) were arthropods, and the remaining 24% (35/147) were plants. Overall, Araneae (30 spiders) and Lepidoptera (44 moths and butterflies) represented a majority total food items detected (Figure 1), but the most common food items detected across fecal sacs were Ericales plants (heather and allies; most commonly blueberries in our study), which were detected in 51% of the fecal sacs.”; lines 288 – 305) and referenced the supplemental file that contains all of the specific information on the analyses.

4.In the discussion the authors refer to only the detected food items and not to the relative abundances of the food items that can shed much more light on the diets especially during winter and non-winter and adults vs nestlings. It can even shed more light on cached vs fresh food items. Through the use of FOO and RRA biases could be explored as well as the possibility of using cached items.

Author response: We agree with the reviewer’s comment that abundance information would be valuable for assessing diet. However, the reason that we have not discussed this in detail is that none of the methods, with the exception of stable isotopes, provide information on abundance. And, because the stable isotope analysis did not strongly agree with the other methods, we are reluctant to discuss the results of this method in the context of abundance. Our Discussion also now includes Table 1 that we hope will help readers compare the limitations of, and the difficulties involved with, different methods of assessing diet. 

5.In the Abstract no reference is made to the type of food items (plants, invertebrates) or how many taxa were detected. There is no mention of dmetabarcoding methods which forms a critical part of the study.

Author response: As suggested, we have now mentioned the types of food items considered in our study and how many taxa were detected. We briefly mention DNA metabarcoding but cannot elaborate further due to word count restrictions.

Specific comments (from attached file)

Line 53: ‘the’ missing

Author response: As suggested, we have included ‘the’ in this statement (line 46).

Line 59: ‘and’ missing

Author response: As suggested, ‘and’ has been inserted (line 59).

Line 77: parentheses needed

Author response: As suggested, parentheses have been included (line 77).

Line 137: More information is needed on the PCR, the PCR protocol followed, enzyme used

Author response: We have significantly added to the methods section and included all of the details surrounding the PCR amplifications for the molecular work. This includes all reagent mixes/volumes, the number and order of the PCRs, and the reaction conditions of the PCR reactions (lines 152 – 187).

Line 152: Sequencing method? By whom was the sequencing done?

Author response: Thank you, the sequencing method and specifics have been included in an updated methods section, including the location of sequencing at the University of Guelph Centre for Biodiversity Genomics (lines 167 – 187).

Line 155: Which package was used to do the quality filtering or did the authors rely on the MULTIQC reports? Which package was used to remove the primers and adapters?

Author response: Both of these methods are now included in the updated methods section. Cutadapt was used to trim sequences and is referenced in the methods (lines 190 – 191). The quality filtering was completed using the fastq quality scores and the sickle tool and the citation is included in the updated methods (lines 189 – 201).

Line 156: Were both the forward and reverse reads used and merged before collapsing identical sequences? Which packages were used to do the bioinformatics in? Were OTUs or ESVs used?

Author response: The sequence reads were generated using Ion Torrent. The data were not reduced to OTU/ESV’s. We have updated the methods section to provide more clarity about the process of analysing the data and include references to the informatic tools used. We have included the sentence “No clustering of the data occurred and all quality filtered reads were used in the taxonomic identification step.” (lines 197 – 198).

Line 160: The methods used to assign taxonomy is unclear.

Author response: The taxonomic assignment was determined using a BLAST approach against focused libraries and where the top BLAST hit was used for taxonomic placement. We have updated the methods section and believe that this question is reflected in these updates (lines 194 – 197). Revised text reads: After these steps, each read was taxonomically assigned using BLAST and three databases: all flowering plant rbcLa sequences from BOLD, all insect COI sequences from GenBank, and all fungi ITS2 sequences from BOLD (Oct 2017).

Line 163: Is there a table containing the species list of the custom database available? How was it obtained? According to which sequence criteria were the sequences for the custom databases acquired?

Author response: We have provided more information on the contents of the databases used to BLAST the experimental sequences against in the updated methods section (lines 202 – 230). Description of the library datasets included noting the taxonomic focus with the following sentence: “After these steps each read was taxonomically assigned using BLAST and three databases including all flowering plant rbcLa sequences from BOLD, all insect COI sequences from GenBank, and all fungi ITS sequences from BOLD (Oct 2017).” (lines 194 – 197)

Line 184: Using which model?

Author response: Yes, we agree that this needed some clarification. There was no model of molecular evolution used. We applied simple distance and to clarify this we have updated the sentence to include the phrase “, the proportion or the number of sites that differ between each pair of aligned sequences,” (lines 224 – 225)

Line 187: This data was not available to me. Only two tables were available as supplemental data.

Author response: Supplemental data have been included in the revised submission.

Line 242: How many raw reads were obtained? Were the sequences obtained deposited onto a database?

Author response: Thank you, for clarification we have significantly updated the methods section for the metabarcoding and we have added several sentences to the beginning of the Results section to clarify the general results of the metabarcode analyses (lines 288 – 305).

Line 248: The results reported on the sequencing are incomplete, and the reference custom databases are not discussed or shown. No results are reported for the gap analysis. Results on the identification of the DNA barcodes are also limited. There is no information added to the table indicating the acronyms for the methods used.

Author response: Thank you for the comment. We have updated the methods section to include all steps taken during the sequencing and taxonomic identification steps (revised text: “To assess if a barcode gap exists for the BLAST identifications, higher taxonomic levels (family for plants and either family or genus for animals depending on the manageable size of the data set and the number of BLAST identified species from the same genus or family) were used to obtain sequence data sets from the BOLD system (Manually downloaded July and August 2018 and unique identifiers are included in the supplemental file S1). Sequences were aligned (MAFFT: 31) and trimmed to target region using MEGA (Using the primers included in Supplemental file S2: 34). Sequences were removed from further analysis if they had greater than 2% unknown nucleotides or if they had greater than 12 gap characters (‘-‘) at either the 3’ or 5’ ends of the sequences. A simple distance, the proportion or the number of sites that differ between each pair of aligned sequences, matrix was constructed with the R package Ape (Ver. 4.1) dist.dna() matrix function (35). These matrices were then used to obtain the maximum within species genetic variation and the minimum genetic distance between species of the same genus and this was completed through a custom R script (see Supplemental Data S3). All within and between taxa values used to assess the specificity of the taxonomic assignments are reported (Supplemental File S4).” lines 216 – 230). 

Lines 261 - 263: Give examples of families or genera detected.

Author response: As suggested, we have included examples of families and genera detected (revised text: “After collapsing taxonomic assignments and assessing the taxonomic placement, 147 unique taxa were obtained as likely taxa from the nestling fecal sacs, 2% (3/147) were vertebrates (wood frog; Lithobates sylvaticus and a shrew; Sorex cinereus), 74% (109/147) were arthropods, and the remaining 24% (35/147) were plants. Overall, Araneae (30 spiders) and Lepidoptera (44 moths and butterflies) represented a majority total food items detected (Figure 1), but the most common food items detected across fecal sacs were Ericales plants (heather and allies; most commonly blueberries in our study), which were detected in 51% of the fecal sacs. The taxonomic identifications of all consumed prey items are provided in Supplementary Table S1.”, lines 297 – 305).

Line 304: I am of the opinion that it will enhance it and not merely duplicate.

Author response: We agree with this comment and have highlighted the benefit of using dDNA metabarcoding below this point in our discussion (“As a practical matter, however, we found no examples in our study where this potential handicap detracted from dDNA metabarcoding’s otherwise superior ability to identify taxa ingested by Canada jays and to determine which items had been “likely cached”. Not only does metabarcoding permit the identification of soft-tissue or small food items that are easily missed by stomach contents analysis (44–46), but also, compared to stomach contents analysis, it allows far more samples to be collected, far more easily, and importantly, without sacrificing individuals. In just three seasons (2015 - 2017) of measuring nestlings, we were able to double the number of identified nestling diet items that we had obtained during the previous 40 years through chance acquisitions of stomach contents and castings. A further advantageous feature is that the technique could be extended to more accurately assess the diets of not just nestling Canada jays, but also those of adults, including the winter diets that are of greatest interest. We have no doubt that it would be eminently feasible to capture winter adults and safely hold them in a suitably appointed small cage until they produced a fecal sample. By the same reasoning, it should be possible, in principle, to use fecal dDNA metabarcoding far beyond the limited use we have made of it here and we encourage others to do so.” Lines 384 – 399).

Reviewer #4: I would like to thank the authors and the editor to give me the opportunity to read the manuscript titled “Fecal DNA metabarcoding helps characterize Canada jay diet and confirms the reliance of stored food for winter survival and breeding”. I found the manuscript very interesting and I think the study has merit, especially in combining molecular and non-molecular techniques.

Author response: We are glad to hear that the reviewer enjoyed reading our manuscript and thank them for their valuable comments that have improved the manuscript.

My expertise is mostly revolving around DNA metabarcoding, so I focused my review on this technique. Unfortunately, I am afraid the manuscript in its current form requires a major revision since most of the information on the use of metabarcoding are missing or are incomplete.

However, I strongly encourage the authors to address my comments and concerns and provide a revised version of their work, since it will be an important tool for future research.

Author response: As suggested, we have revised the methods section of our manuscript to provide additional detail that was missing or incomplete in our initial submission. We provide specific responses, including highlighting text sections which have changed and line numbers, to your comments below.

Main comments:

If the ITS marker could not be amplified (despite following published protocols), the authors should probably remove any reference to it everywhere in this manuscript.

Author response: Despite not being able to amplify the ITS marker, we think it is important to say that this was attempted as fungi could be an important part of the Canada jay diet and future studies could attempt to conduct this analysis.

Metabarcoding methods needs to be improved:

Page 7: No information is provided for the sequencing. I think it is very important to provide this kind of information in detail:

- What did the authors do after PCR amplification?

- How did they prepare the sequencing library?

- Were the samples run in duplicate?

- Were there controls to account for plant/fungi present in the nest but not part of the diet?

- Were dual unique indexes used?

- Who performed the sequencing (in-house or outsourced)?

- What platform did the authors used to perform the sequencing?

- Were all markers pooled in the same sequencing run?

Author response: Thank you. We have included an updated methods section as per suggestions from reviewers. In this section we believe that we have addressed all of the concerns noted above with the lack of methodological specifics (lines 128 – 187).

Page 8: Metabarcoding reads quality control is not really mentioned, except for size and primer trimming. Were there any chimera removal? Were coding genes checked for stop codons? Especially for fungi, was there any threshold-based OTU clustering?

Author response: Thank you for the questions. This work utilized unidirectional Ion Torrent sequencing so there was no need to merge reads. With no merging there was also no need to check for chimeric sequences. We did not employ checks for the coding markers. And there was no clustering that was completed. We have included clarifying language for the instrument and lack of clustering in the updated methods section (lines 184 – 198). Revised text reads: “Cleaned and normalized products used for library construction following Prosser and Hebert (23) and were then sequenced unidirectionally on an Ion Torrent PGM using a 318 v.2 chip at the University of Guelph Centre for Biodiversity Genomics, following the manufacturer’s instructions. 

Bioinformatics

Sequencing data was demultiplexed using the gene region and MID tags. Each resulting data set (three gene regions for each sample) were analysed informatically by first removing primer and adapter sequences (31) ; see Prosser and Hebert (23) for sequences used), removing reads shorter than 100 bp, removing reads with low quality scores (QV < 20) (github.com/ucdavis-bioinformatics/sickle), and then by de-replicated reads with 100% identity (http://hannonlab.cshl.edu/fastx_toolkit/index.html). After these steps, each read was taxonomically assigned using BLAST and three databases: all flowering plant rbcLa sequences from BOLD, all insect COI sequences from GenBank, and all fungi ITS2 sequences from BOLD (Oct 2017). No clustering of the data occurred, and all trimmed and quality filtered reads were used in the taxonomic identification step.”

Page 9: please include software references for R/RStudio and package version for each of the packages used.

Author response: As suggested, we have made sure to include citations for all software and R packages used in our analyses (lines 222 – 230).

Results:

The authors should summarise the fate of the raw reads obtained using metabarcoding, ideally for each genetic marker. For example:

“A total of XX millions raw reads was generated from XX samples. Of these, X thousand reads passed quality controls. A total of XX taxa could be identified based on COI, XXX taxa for ITS, XXX taxa for Rbcl”

Author response: As suggested, we have added this information to the results section (lines 228 – 306). We have included a number of sentences detailing the outcomes of each of our analysis steps starting at the beginning of the results section. The revised text now reads: “Metabarcoding results from the fledgling faecal samples were represented by 20 biological collections representing three gene regions. Of the three gene regions, raw sequencing results for the ITS2 focusing on fungal taxa had between 0 and 6349 reads, rbcLa focusing on the plant taxa had between 5 and 852,304 reads, and COI-5P focusing on animal taxa had between 0 and 267,578 reads for a total of 3,761,841 (Supplemental file S5). After trimming and filtering there were 2,775,314 reads across 20 samples representing 1469 unique sequences, 1092 from the COI-5P, 370 from the rbcLa, and 7 ITS2 (Supplemental file S6).” (lines 288 – 294).

Raw data for metabarcoding analysis should be made publicly available.

Author response: We agree and these data will be made publicly available once the manuscript is accepted.

Minor comments:

Line 132: Change “molecular regions” to “gene regions”.

Author response: As suggested, we have changed ‘molecular regions’ to ‘gene regions’ (line 129).

Line 133: Italicise the name of the genes when reported in full. “Cytochrome c Oxidase” should be “Cytochrome c Oxidase” and so on.

Author response: As suggested, all gene names have been italicized when reported in full (lines 130 – 133). 

Line 139: the expression “molecular gene region” is used a number of times. I think the use of “molecular” in this case is redundant since any gene region is “molecular” in a way. Can you please remove all instances of “molecular” and leave only “gene region”? At line 138, “molecular marker” is correct and can be used since it specify the type of marker.

Author response: As suggested, we have changed all references to ‘molecular gene region’ to gene region’.

Lines 144-145: repetition of “well established” twice over two sentences. If you could remove one instance, it may make the sentence flow better.

Author response: As suggested, we have removed the first ‘well established’ from this statement (lines 140 – 143).

Line 147: Change “molecular regions” to “gene regions”.

Author response: As suggested, we have changed ‘molecular region’ to ‘gene region’.

Line 155: This sentence is missing something “Primers were trimmed from each read and filtered again by a minimum size of 100 bp”. The authors trimmed the primers and then filtered. the reads? As it stands, the subject of the second sentence is still “primers”.

Author response: As suggested, we have updated the methods section and clarified these points (lines 189 – 194). Revised text reads: “Sequencing data was demultiplexed using the gene region and MID tags. Each resulting data set (three gene regions for each sample) were analysed informatically by first removing primer and adapter sequences (31) ; see Prosser and Hebert (23) for sequences used), removing reads shorter than 100 bp, removing reads with low quality scores (QV < 20) (github.com/ucdavis-bioinformatics/sickle), and then by de-replicated reads with 100% identity (http://hannonlab.cshl.edu/fastx_toolkit/index.html).”

Line 244 (and line 226): Following the rules of the International Code of Zoological Nomenclature, only names of genera and species should be italicised, but not families and orders. Araneae and Lepidoptera (and Zapodidae) should not be italicised. Also the abbreviation “sp.” should not be italicised (line 227).

Author response: As suggested, families are no longer italicized (lines 299 – 303).

Line 246: I am less familiar with plant taxonomy, but I think the above rule also applies there. Ericales should not be italicised.

Author response: As suggested, Ericales is no longer italicized (line 302). 

Line 247: no need of upper case for “blueberries”.

Author response: As suggested, blueberry is no longer capitalized (line 303).

Reviewer #5: This is a very interesting work, and only a few issues need the author's attention.

Author response: We thank the reviewer for positive assessment of the manuscript and for their helpful comments.

1. More literature on DNA fecal metabarcoding needs to be added in the introduction section.

Author response: As suggested, we have provided additional literature to the introduction.

2. The methods section needs to be greatly simplified, especially the section on sequencing and bioinformatics. In addition, it is necessary to indicate which sequencing platform was used for sequencing.

Author response: Several other reviewers asked for more information in this section. We have included details requested by other reviewers while trying to ensure the methods are as simplified as possible.

3. What does "nestlings" mean？How did this research get samples of nestlings？Do young birds at different stages of development need to be distinguished?

Author response: Nestling is a widely accepted term in ornithological research that describes a developing individual in a nest. While developmental stages could be differentiated, we do not believe this level of detail is necessary in our analysis.

4. Why not use ITS2 to amplify plants, as suggested in this paper “Validation of the ITS2 region as a novel DNA barcode for identifying medicinal plant species”?

Author response: Thank you. We have provided more information in the discussion about the choices made for our methods and potential other options for studies in the future. To address your specific point regarding the ITS2 gene region and identifying plant species we have included the following sentence in the discussion. “For example, while the ITS2 region was amplified and sequenced for use in fungal identification (although unsuccessfully most likely due to the degree of degradation occurring for these soft bodied organisms), the ITS region has been successfully utilized for identification of other taxa including Canadian flora (43). However, due to the time and cost constraints to amplify the gene region using additional primers, in addition to the poorly populated sequence libraries for this taxonomic group, this approach was not feasible for this study.” (lines 371 – 377).

5. Could you consider using "shotgun metabarcoding" for further research? At least the relevant discussion should be added to the discussion section. Relevant literature can be used for reference, for example “The species identification in traditional herbal patent medicine, Wuhu San, based on shotgun metabarcoding”.

Author response: Thank you for the suggestion. We have included the following sentence in the discussion section to highlight the possibility. “One possible solution to primers poorly amplifying taxa, or alternatively, primers preferentially amplifying taxa, is to take a PCR-free approach and, instead, obtain sequences for all available DNA in the samples. This is sometimes referred to as a shotgun metagenomics sequencing and while this approach may help to address some primer concerns and could offer potential for prey biomass estimates it is not without challenges in application, including high costs and the generation of large datasets leading to computationally intensive analyses (41, 42).” (lines 361 – 366).

---

## [Decision Letter · Decision Letter 1]

26 Dec 2023

PONE-D-23-24828R1Fecal DNA metabarcoding helps characterize the Canada jay’s diet and confirms its reliance on stored food for winter survival and breedingPLOS ONE

Dear Dr. Sutton,

Thank you for submitting your manuscript to PLOS ONE. After careful consideration, we feel that it has merit but does not fully meet PLOS ONE’s publication criteria as it currently stands. Therefore, we invite you to submit a revised version of the manuscript that addresses the points raised during the review process.

We look forward to receiving your revised manuscript.

Kind regards,

Petr Heneberg

Academic Editor

PLOS ONE

Journal Requirements:

Reviewers' comments:

Reviewer's Responses to Questions

**Comments to the Author**

1. If the authors have adequately addressed your comments raised in a previous round of review and you feel that this manuscript is now acceptable for publication, you may indicate that here to bypass the “Comments to the Author” section, enter your conflict of interest statement in the “Confidential to Editor” section, and submit your "Accept" recommendation.

Reviewer #2: All comments have been addressed

Reviewer #3: All comments have been addressed

Reviewer #4: All comments have been addressed

2. Is the manuscript technically sound, and do the data support the conclusions?

Reviewer #2: Yes

Reviewer #3: Yes

Reviewer #4: Yes

3. Has the statistical analysis been performed appropriately and rigorously? 

Reviewer #2: Yes

Reviewer #3: N/A

Reviewer #4: Yes

4. Have the authors made all data underlying the findings in their manuscript fully available?

Reviewer #2: Yes

Reviewer #3: Yes

Reviewer #4: Yes

5. Is the manuscript presented in an intelligible fashion and written in standard English?

Reviewer #2: Yes

Reviewer #3: Yes

Reviewer #4: Yes

6. Review Comments to the Author

Reviewer #2: My main concerns have been addressed and I am satisfied with the answers and corrections.

Methods are now described in detail.

One note is that if you were following standard kits (i.e. for DNA extraction) and followed manufacturers instructions then you can skip giving all the details and just say "we used this-and-that kit following manufacturers instructions".

Small comments:

general: Degrees symbol doesn't appear well in the PDF. To be fixed in formatting.

general: Sometimes you write "COI" and sometimes "COI-5P". I'm not sure what the difference is. Usually "COI" is sufficient.

Line 126-127: Thanks for clarifying. In this case you can just write that it was stored in -20 degrees within 12hrs of collection and until further processing.

Line 130: You should write full name of the region first - Cytochrome c Oxidase I - and abbreviation in parentheses. Also only "c" should be italicised. Check it throughout the text.

Line 131: Same with ITS. First write the full name of the fragment and then add abbreviation in parentheses.

Line 132: same

Line 224: Thanks for clarifying what the distance metric is. Now I think that "simple" is necessary. This is not a precise word. Just delete it and rephrase.

Reviewer #3: I want to commend the authors for the effort they put in to improve the manuscript and the attention they paid to reviewers’ comments. The authors made major changes to the manuscript that improved the value of the manuscript considerably. I like to recommend that the authors also consider the UNITE database for fungal sequence identification in possible future studies using DNA metabarcoding. I have noticed a few minor errors in the methods section:

Line 173: 10 µM (not lM) primer, 0.0625 μL of 10 mM dNTP (KAPA Biosystems), 0.060 µL (not lL) of 5U/μL PlatinumTaq

Line 175: 175 reactions consisted of 94 °C for 5 min, 40–60 cycles (40 for ITS2, 60 for rbcLa (not A) and COI)

Reviewer #4: I am very happy with the answers and changes provided by the authors.

I think the manuscript is now ready to be accepted.

I have only two very minor remarks:

- Lines 129-145: the paragraph on the selection of the markers should go with the PCR section, not before the DNA extraction. DNA extraction and purification happen before PCR, and they should be reported in this order.

- Check the whole manuscript for the use of the degree symbol °C. Not sure what symbol was used instead, but it really looks weird.

Great work, looking forward to see this published! Congratulations!

7. PLOS authors have the option to publish the peer review history of their article (what does this mean?). If published, this will include your full peer review and any attached files.

Reviewer #2: No

Reviewer #3: No

Reviewer #4: No

---

## [Author Response · Author response to Decision Letter 1]

27 Feb 2024

We thank the reviewers for their comments on our revised manuscript. Our responses to their comments are highlighted in the manuscript in red and bolded below for clarity.

Reviewer #2: My main concerns have been addressed and I am satisfied with the answers and corrections.

Methods are now described in detail.

One note is that if you were following standard kits (i.e. for DNA extraction) and followed manufacturers instructions then you can skip giving all the details and just say "we used this-and-that kit following manufacturers instructions".

Author response: Thank you for your suggestion. We have left the description of our manual extraction methods in the manuscript because it describes the methods followed throughout the analysis.

Small comments:

general: Degrees symbol doesn't appear well in the PDF. To be fixed in formatting.

Author response: As suggested, we have made sure all degree symbols are now being displayed appropriately.

general: Sometimes you write "COI" and sometimes "COI-5P". I'm not sure what the difference is. Usually "COI" is sufficient.

Author response: There is a difference between COI and COI-5P. For consistency and to reflect the fact we used the COI-5P, we have made sure all mention of COI throughout the manuscript has been changed to COI-5P.

Line 126-127: Thanks for clarifying. In this case you can just write that it was stored in -20 degrees within 12hrs of collection and until further processing.

Author response: As suggested, we have revised the text to state this.

Line 130: You should write full name of the region first - Cytochrome c Oxidase I - and abbreviation in parentheses. Also only "c" should be italicised. Check it throughout the text.

Author response: As suggested, we have made sure to write the full name first and only italicize the “c”.

Line 131: Same with ITS. First write the full name of the fragment and then add abbreviation in parentheses.

Author response: As suggested, we have made sure to write the full name first. 

Line 132: same

Author response: As suggested, we have made sure to write the full name first. 

Line 224: Thanks for clarifying what the distance metric is. Now I think that "simple" is necessary. This is not a precise word. Just delete it and rephrase.

Author response: As suggested, we have removed the word “simple” and revised the sentence to describe the matrix that was created. The revised text now reads: “A distance matrix (the proportion or the number of sites that differ between each pair of aligned sequences) was constructed with the R package Ape (Ver. 4.1) dist.dna() matrix function (35).” (lines 224 – 226)

Reviewer #3: I want to commend the authors for the effort they put in to improve the manuscript and the attention they paid to reviewers’ comments. The authors made major changes to the manuscript that improved the value of the manuscript considerably. I like to recommend that the authors also consider the UNITE database for fungal sequence identification in possible future studies using DNA metabarcoding. I have noticed a few minor errors in the methods section:

Line 173: 10 µM (not lM) primer, 0.0625 μL of 10 mM dNTP (KAPA Biosystems), 0.060 µL (not lL) of 5U/μL PlatinumTaq

Author response: Thank you for catching these mistakes. We have revised the text to correct these errors. 

Line 175: 175 reactions consisted of 94 °C for 5 min, 40–60 cycles (40 for ITS2, 60 for rbcLa (not A) and COI)

Author response: We now have removed the [] from the sentence.

Reviewer #4: I am very happy with the answers and changes provided by the authors.

I think the manuscript is now ready to be accepted.

I have only two very minor remarks:

- Lines 129-145: the paragraph on the selection of the markers should go with the PCR section, not before the DNA extraction. DNA extraction and purification happen before PCR, and they should be reported in this order.

Author response: As suggested, we have moved this paragraph to be below our description of the PCR method used and now appears on lines 177 – 187. 

- Check the whole manuscript for the use of the degree symbol °C. Not sure what symbol was used instead, but it really looks weird.

Author response: We have made sure all uses of °C are now consistently formatted.

Great work, looking forward to see this published! Congratulations!

Author response: Thank you for the kind words and for your helpful comment on previous versions of the MS.

---

## [Editor Report · Decision Letter 2]

1 Mar 2024

Fecal DNA metabarcoding helps characterize the Canada jay’s diet and confirms its reliance on stored food for winter survival and breeding

PONE-D-23-24828R2

Dear Dr. Sutton,

We’re pleased to inform you that your manuscript has been judged scientifically suitable for publication and will be formally accepted for publication once it meets all outstanding technical requirements.

Kind regards,

Petr Heneberg

Academic Editor

PLOS ONE
---

## [Editor Report · Acceptance letter]

2 Apr 2024

PONE-D-23-24828R2 

PLOS ONE

Dear Dr. Sutton, 

I'm pleased to inform you that your manuscript has been deemed suitable for publication in PLOS ONE. Congratulations! Your manuscript is now being handed over to our production team.

Kind regards, 

on behalf of

Dr. Petr Heneberg 

Academic Editor

PLOS ONE